# An Insight into the Prospects and Drawbacks of Stem Cell Therapy for Spinal Cord Injuries: Ongoing Trials and Future Directions

**DOI:** 10.3390/brainsci13121697

**Published:** 2023-12-09

**Authors:** Shahidul Islam Khan, Nazmin Ahmed, Kamrul Ahsan, Mahmud Abbasi, Rosario Maugeri, Dhiman Chowdhury, Lapo Bonosi, Lara Brunasso, Roberta Costanzo, Domenico Gerardo Iacopino, Giuseppe Emmanuele Umana, Bipin Chaurasia

**Affiliations:** 1Spine Surgery Unit, Department of Orthopaedic Surgery, Bangabandhu Sheikh Mujib Medical University, Shahbag, Dhaka 1000, Bangladesh; sikhan55@yahoo.com (S.I.K.); kahsansps@yahoo.com (K.A.); 2Department of Neurosurgery, Ibrahim Cardiac Hospital and Research Institute, Shahbag, Dhaka 1000, Bangladesh; nazmin.bsmmu@gmail.com; 3Department of General Anaesthesiology, Ibrahim Cardiac Hospital and Research Institute, Shahbag, Dhaka 1000, Bangladesh; drmahmudabbasi8@gmail.com; 4Neurosurgical Clinic, AOUP “Paolo Giaccone”, Post Graduate Residency Program in Neurologic Surgery, Department of Experimental Biomedicine and Clinical Neurosciences, School of Medicine, University of Palermo, 90133 Palermo, Italy; lapo.bonosi@community.unipa.it (L.B.); lara.brunasso@community.unipa.it (L.B.); roberta.costanzo@community.unipa.it (R.C.); gerardo.iacopino@gmail.com (D.G.I.); 5Department of Neurosurgery, Bangabandhu Sheikh Mujib Medical University, Shahbag, Dhaka 1000, Bangladesh; dhiman_ns@yahoo.com; 6Department of Neurosurgery, Trauma Center, Gamma Knife Center, Cannizzaro Hospital, 95126 Catania, Italy; umana.nch@gmail.com; 7Department of Neurosurgery, Neurosurgery Clinic, Birgunj 44300, Nepal; trozexa@gmail.com

**Keywords:** spinal cord injury, stem cells, stem cell therapy, clinical trials

## Abstract

Spinal cord injury (SCI) is a devastating neurological disorder that has a substantial detrimental impact on a person’s quality of life. The estimated global incidence of SCI is 40 to 80 cases per million people and around 90% of cases are traumatic. Various etiologies can be recognized for SCI, and post-traumatic SCI represents the most common of these. Patients worldwide with SCI suffer from a persistent loss of motor and sensory function, which affects every aspect of their personal and social lives. Given the lack of effective treatments, many efforts have been made to seek a cure for this condition. In recent years, thanks to their ability to regenerate tissue and repair lost or damaged cells, much attention has been directed toward the use of stem cells (embryonic, induced pluripotent, mesenchymal, hematopoietic), aimed at restoring the functional integrity of the damaged spinal cord and improving a functional recovery including sensory and motor function. In this paper, we offer an overview of the benefits and drawbacks of stem cell therapy for SCI based on clinical evidence. This report also addresses the characteristics of various stem cell treatments, as well as the field’s likely future. Each cell type targets specific pathological characteristics associated with SCI and demonstrates therapeutic effects via cell replacement, nutritional support, scaffolds, and immunomodulation pathways. SCI accompanied by complex pathological processes cannot be resolved by single treatment measures. Stem cells are associated with the adjustment of the expression of neurotrophic factors that help to achieve better nutrition to damaged tissue. Single-cell treatments have been shown in some studies to provide very minor benefits for SCI in multiple preclinical studies and a growing number of clinical trials. However, SCI damage is complex, and many studies are increasingly recognizing a combination approach such as physical therapy, electrical stimulation, or medication therapy to treatment.

## 1. Introduction

Spinal cord injury (SCI) represents a significant financial and psychological burden, including anxiety, as well as excessive fear regarding patients’ wellbeing and healthcare systems [1,2]. The etiology can be varied, but, generally, SCI can be divided into traumatic and non-traumatic causes. Traumatic SCI is more common (up to 90% of cases) and is typically caused by external physical impacts. Tumor compression, vascular ischemia, or congenital disease are common causes of non-traumatic SCI. Trauma from accidents, falls, gunfire, or medical/surgical complications are the most common causes. SCI remains a syndrome that primarily affects young people due to the nature of its causes. However, living in an aging society has led to an increase in the frequency of new occurrences of SCI in the older population following low-energy trauma. Pathophysiological events that occur after an injury frequently result in lasting neurological deficits, such as the loss of motor and sensory function below the damage level, as well as autonomic dysfunction. From a pathophysiological point of view, cell death, axonal collapse and demyelination, glial scar formation, and abnormal and sustained inflammation are the main mechanisms at the base of SCI [3,4]. To date, there are few effective therapies for SCI, but no cure is known. The currently available options include spinal cord decompression surgery, as well as medical and physical therapy; however, none of them result in a marked improvement in function, especially when the damage has already established. Individual treatments are inadequate to elicit neural regeneration and functional recovery following SCI. In this context, effective and safe regenerative strategies that promote spinal cord repair need to be found. In recent decades, different regenerative strategies have been suggested, including direct cell transplantation, growth factor injections, and tissue engineering strategies based on the combination of biomaterial, stem cells, and growth factors [5,6]. 

The damage mechanism in traumatic SCI presents two stages. The initial traumatic impact on the spinal cord causes microhemorrhages in the white and gray matter. Following the primary injury, a cascade of pathophysiological events results in altered neuronal homeostasis, apoptosis, and tissue destruction. SCI is pathophysiologically grouped into primary and secondary injuries and where the acute phase timeline is <48 h, the subacute is between 48 h to 14 days, the intermediate phase is between 14 days to 6 months, and the chronic phase is >6 months [7,8,9]. In this context, the current clinical practice focuses on surgical decompression (with no clear indication about the correct timing for surgical decompression) and, eventually, mechanical stabilization with rods and screws, followed by a pharmacological intervention such as high-dose methylprednisolone. Some controversy has arisen in this regard, and a subsequent analysis of the National Acute Spinal Cord Injury Study (NASCIS) II and III studies demonstrated potentially serious complications from intravenous methylprednisolone with limited benefits, so its use is debated. Therefore, dexamethasone has been widely considered as an alternative and is also under investigation [8,10]. In addition to surgical and pharmacological therapies, a key role to regain patients’ functionality and autonomy is played by physical and rehabilitative strategies, including occupational therapy, robotic rehabilitation, functional electric stimulation, muscle and locomotor training, etc. Unfortunately, all these interventions have demonstrated poor outcomes regarding neuroprotection, neuroregeneration, and functional recovery. The reason behind this failure lies in the complexity of the pathophysiological mechanisms of SCI, which result in irreversible damage to the neuronal environment at the site of the injury. The time-sensitive and complex pathophysiology makes it particularly difficult to investigate the therapeutic targets for SCI [9,11,12].

Furthermore, individuals with SCI frequently face socioeconomic issues, particularly if they live in countries with limited possibilities and social assistance for disabled persons. The discrimination in access to care rehabilitation services indicates the need for professionals in this area to advocate for the tailoring of social support to reduce disparities. In recent decades, in an effort to find potential therapeutic solutions, stem cell therapy has emerged as a very promising approach in the field of SCI. In light of results from trials about promising therapy in animals, clinical trials involving humans with SCI have started and became a reality in the middle of the first decade of this century [13,14,15]. This scoping review aims to provide an overview of the benefits and drawbacks of various stem cells in SCI and address the characteristics of numerous stem cell treatments, the main results of related clinical trials, and the future research scope in this interesting and promising field. 

## 2. Methodology

Inclusion and exclusion criteria

For this review, we have included clinical trials with all types of study design with the following criteria: completed clinical trials or ongoing clinical trials with a protocol, having a specific and clear target, and trials for evaluating stem cell therapy in spinal cord injury. We have excluded animal trials and unregistered studies. 

The website ClinicalTrials.gov was used to search relevant articles. The language limitation was set English and the search deadline was December 2021. 

## 3. Ongoing Clinical Trials of Stem Cell Therapy for SCI

The number of clinical trials based on stem cells has increased in recent years. There are already thousands of registered studies worldwide that claim to use “stem cells” in experimental treatments [16]. Several clinical trials are ongoing throughout the world to investigate the safety and efficacy of stem cell treatment for SCI, and they use a range of stem cell sources and delivery modalities. The cell preparation and transplantation technique in clinical trial research should follow or correspond to the norms and guidelines. Clinical study results will aid in determining the best type of stem cell to use, as well as the most successful method of administration, and prospective, multicenter, double-blind or observing-blind, placebo-control, randomized control trials need to be conducted to firmly prove the neurorestorative effects or obtain greater benefits. With continued biotechnological and nanomaterial advances, stem cell therapy for the treatment of SCI can become a real therapeutic option over the next few years, and focus on the molecular mechanisms related to both the injury and the processes of functional recovery and regeneration of nerve tissue can bring a paradigm shift in the management of SCI [17]. Table 1 displays a list of clinical trials including stem cell treatment for SCI. 

## 4. Advances and Prospects of Stem Cell Therapy for SCI

Recent advancements in stem cell research have boosted the prospect of novel SCI therapeutics. Stem cells are a type of cell in the body that can differentiate into numerous types of cells. Given their ability to heal injured nervous tissue, they are a prospective candidate for SCI treatment. Generally speaking, they can be classified into two main categories: embryonic stem cells (ESCs) and adult stem cells. ESCs can develop into any form of cell in the body. Adult stem cells, on the other hand, can specialize into specific types of cells and are found in a variety of organs throughout the body, including bone marrow. Stem cells have been found in animal models of SCI to improve motor and sensory function and have also been shown to enhance axon regeneration, reduce inflammation, and increase tissue repair. According to a recent meta-analysis, the ASIA impairment scale score can improve at least in one grade in 48.9% of individuals with SCI; furthermore, it can improve urinary and gastrointestinal system function by 42.1 and 52.0 percent, respectively [17].

Despite the fact that ESCs have the ability to differentiate into any type of cell, their usage is controversial due to ethical considerations. Recent studies have concentrated on induced pluripotent stem cells (iPSCs), which are adult cells that have been reprogrammed to act like ESCs. iPSCs can differentiate into any type of cell and can be created from the patient’s own cells, lowering the chance of rejection. The generation methods of iPSCs vary in the vehicles of genes, combinations of reprogramming factors, and cell types. Yamanaka factors (combination of *OCT3/4*, *SOX2*, *KLF4*, and *C-MYC*) are common techniques; however, the quality would be improved by introducing Zscan4 through forced expression. Furthermore, it has been documented how stem cells produce a variety of growth factors, cytokines, and chemokines that aid in tissue repair and regeneration, reducing the pro-inflammatory environment related to SCI and decreasing scar production. This paracrine impact has been proven in preclinical models of SCI to increase functional recovery and may also contribute to the favorable benefits of stem cell therapy in clinical studies [18,19,20]. 

Mesenchymal stromal cells have more clinical research reported in SCI treatment than other types of cells. Their advantages include abundant sources, as well as easy culture and preparation procedures. A retrospective study observed some functional improvements in one third of patients with acute complete SCI and also in nearly half of patients with chronic complete SCI after a two-to-five-year follow-up. Repeated subarachnoid administrations of umbilical cord MSCs in 41 patients with SCI led to a significant improvement in assessing neurological dysfunction and the quality of life in a phase 1/2 pilot study [18,21]. Vaquero et al. reported autologous bone marrow MSC transplantation in 12 patients with chronic complete paraplegia; all patients experienced functional improvement, including sensitivity, sphincter control, motor activity, decreases in spasms and spasticity, and improved sexual function, significantly improving their overall quality of life. Also, this kind of cell therapy could relieve neuropathic pain due to SCI, reduce syrinx, and show clinical improvements for post-traumatic syringomyelia [22,23,24,25]. 

Zhao et al. [21] implanted scaffolds with umbilical cord MSCs following scar resection with chronic complete SCI; some functional improvements were observed in some patients during 1 year of follow-up. Larocca et al. [26] reported that the image-guided percutaneous intralesional administration of MSCs showed some functional improvement. Santamaría et al. [27] reported that intrathecal injections of bone marrow stromal cells in a patient with C2 tetraplegia were associated with clinical and neurophysiological improvement [26,27,28]. A study conducted by Park et al. [29] observed negligible improvements in the motor power of the upper extremities and in activities after directly injecting autologous bone marrow MSCs into both the spinal cord and the intradural space with SCI. Oraee-Yazdani et al. reported that the transplantation of an autologous MSC and SC combination directly into the injury site showed negligible sensory improvement [30,31,32]. 

Some research also found no motor improvement in patients with chronic SCI who received an intrathecal transplantation of umbilical cord MSCs. Chotivichit et al. [33] used MRI to track autologous bone MSC transplantation in a patient with persistent SCI; however, this showed no functional improvements. According to Satti et al. [34], the intrathecal injection of autologous MSC transplantation is safe in individuals with full SCI [35].

Basic research has also shown that hematopoietic stem cells can help with SCI. HSCs are advantageous due to their ability to be autologously-derived and have a record of safety in humans, but they are rare and pose major risks with graft rejection. In a retrospective investigation, Al-Zoubi et al. [36] implanted purified autologous leukapheresis-derived CD34+ and CD133+ stem cells into 19 patients with chronic full SCI; over half of the patients showed segmental sensory or motor improvement after long-term follow-up. Ammar et al. [37] revealed in a pilot trial, using a biological scaffold containing autologous HSCs and platelet-rich protein for four individuals with SCI, a patient who demonstrated motor and objective sensory improvement.

Some prior research found a slight neurological improvement from transplanting neural stem/progenitor cells into people with chronic SCI. NSCs are multipotent and can replace damaged neural tissue and have the capacity for neuronal differentiation and functional improvement. In a phase I trial, perilesional intramedullary injections of human central nervous system stem cells (HuCNS-SC) from a fetal brain proved safe and viable; however, a few patients demonstrated small motor benefits in phase II single-blind, randomized research. Transplanting HuCNS-SCs into chronic SCI patients’ injured thoracic cords revealed safety, and follow-up studies indicated consistent sensory improvements without motor improvement [29,38,39,40,41]. 

## 5. Drawbacks of Stem Cell Therapy for SCI

One of the most complex challenges in the realm of stem cell therapy for SCI is the careful selection of the most suitable type of stem cells. A substantial hurdle is the lack of viable stem cell sources. Though it is possible to generate induced pluripotent stem cells (iPSCs) from adult cells, this approach comes with an elevated risk of cancer development. Furthermore, the delivery of stem cells to the injured area is an obstacle that should not be underestimated. Stem cells must be introduced in a manner that ensures their survival and integration with the host tissue. This is particularly difficult due to the harsh environment present at the injury site, characterized by inflammation and the presence of inhibitory chemicals that can interfere with cell survival and differentiation. Allogeneic stem cells, such as embryonic stem cells (ESCs) or iPSCs, when transplanted into a recipient, can elicit an immunological response. To mitigate this, strategies like immunosuppressive medications and the utilization of autologous stem cells are employed to reduce the risk of immunological rejection. Previous research has documented 28 distinct adverse effects associated with stem cell transplantation. These include neuropathic pain, unusual sensations, muscle spasms, vomiting, and urinary tract infections. However, it is worth underlining that major side effects such as cancer may not have been observed due to relatively short follow-up periods [17]. Assessing the effectiveness of stem cells through head-to-head comparisons in meta-analyses is challenging. Variations in factors such as cell types, sources, culture conditions, patient demographics, and SCI severity contribute to the complexity of research in this field [30,42].

Existing research has unfortunately only demonstrated minor improvements in sensory and motor function, which do not meet the criteria necessary for walking or performing daily activities. Moreover, the recovery of bowel and bladder function, which many SCI patients consider vital, has shown limited improvement in current studies. Enrolling an adequate number of SCI patients in clinical trials is hindered by criteria such as injury severity, patient age, and overall physical condition. The accuracy of subjective outcome measurements, including the ASIA scale, can be compromised, leading to imprecise assessments of stem cell efficacy [21,28,43,44].

The potential for spontaneous recovery and the lack of a control group in most trials make it difficult to conclusively determine whether the observed therapeutic effects are solely attributable to stem cell transplantation [17,45,46]. Further research is imperative to delve deeper into the actual therapeutic effects of stem cells through standardized controlled trials. The implementation of blinding techniques is crucial to ensure the reliability of clinical trial outcomes. Many prior studies have failed to report adverse events comprehensively, potentially resulting in an overstatement of stem cell safety and efficacy [21,36,37]. Consequently, extending the follow-up period is essential to comprehensively investigate the safety profile of stem cell therapy. 

Additional limitations in stem cell research encompass issues like inadequately estimating patient enrollment numbers for trials and, consequently, therapeutic efficacy being lower than in animal models; inconsistent criteria for patient inclusion and exclusion; variability in stem cell sources and transplantation methods which are different from animal studies; and the lack of standardized risk assessment methods for tumorigenicity and oncogenicity [37,39,47,48,49,50].

## 6. Future Research and Directions

For designs in the context of stem cell therapy for SCI, more stringent testing requirements must be established. The lack of suitable control groups makes it difficult to reach a conclusive determination in most trials; therefore, the incorporation of an appropriate control group in trials will be helpful to reach a conclusion. Future research should prioritize elucidating the specific mechanisms by which stem cells facilitate tissue repair and regeneration; however, studies should follow standardized research methodology to enhance the quality of findings [13,17,51]. 

Improvements in delivery systems are a critical area for future research. The use of iPSCs, which can be derived from a patient’s own cells, holds promise in addressing the scarcity of autologous stem cells. Additionally, more efficient delivery methods, including microscale and nanoscale delivery systems, should be explored to enhance stem cell survival and integration. Techniques such as injection, implantation, and scaffolding all have their respective advantages and disadvantages, necessitating further research to determine the most effective mode of delivery [52,53,54,55].

Optimizing stem cell differentiation and integration into host tissue is another pressing area for future investigation. The application of gene editing technology has the potential to precisely modulate stem cell differentiation and integration. Moreover, tissue engineering technologies, such as organoid production or the creation of bioengineered spinal cord tissue, could provide a more physiologically realistic environment for stem cell differentiation and integration. Interdisciplinary approaches are also important, involving bioengineering and neurology for more precise outcomes [56,57,58].

To address concerns about the safety of stem cell therapy, particularly regarding tumor growth and immunological rejection, ongoing research efforts should focus on developing techniques to mitigate these risks and improve transplantation efficacy. This entails accurate sample size calculations, the inclusion of well-diagnosed SCI patients, and ensuring a sufficiently long follow-up duration to comprehensively assess potential adverse events. The genome has been popular to improve the safety of cell transplantation therapies. Orthogonol systems have also been developed to selectively kill cell products if necessary [59,60].

Exploring combinations of stem cell therapy with other treatments, such as physical therapy, electrical stimulation, or medication therapy, is a promising avenue for future research. Research focusing on the combination of genetic engineering technology with nanobiotechnology, combinational therapy with neuroprotective agents, cell coupling, and rehabilitation may help to improve the effectiveness of stem cells. Rigorous clinical trials and long-term follow-up studies are paramount for advancing stem cell therapy for SCI and, ultimately, enhancing patient outcomes [61,62,63]. There are guidelines from the International Society for Stem Cell Research for “ethical, scientifically, medically and socially responsible” stem cell research. There is also a series of guidelines and standards for clinical trial design from the International Campaign for Cures of Spinal Cord Injury Paralysis (ICCP). However, proper guidelines focused on the correct management of SCI using stem cells, including the identification of the most suitable stem cells, delivery site, and timing of intervention, are required to be established. Effective collaboration among policymakers, researchers, and clinicians is vital for developing appropriate guidelines [17,64].

To overcome the obstacles regarding safety, therapeutic efficacy, and immunocompatibility, extensive research on gene-editing technologies (i.e., CRISPR/Cas9) should be evolved. Advanced research on tissue engineering techniques would be able to identify more effective therapies, growth factors, and bio-active molecules, which could result in a breakthrough in this area. A conductive environment for the survival, differentiation, and integration of stem cells is a challenge, and 3D-printed biometric spinal cords have shown some efficacy; thus, more research is required in this area to design advanced scaffolds and biomaterials. Moreover, research should be conducted to identify new sources of stem cells, which would have better efficacy and safety. Continuous efforts should also be given toward the combination of stem cell therapy with electrical stimulation and personalized therapeutic interventions [17,22,64]. 

## 7. Conclusions

Stem cell therapy has emerged as a potential and valuable treatment option for spinal cord injury, aiming to guarantee cellular and tissue regeneration and improve functional recovery. Stem cells produce a variety of growth factors and are, thus, associated with motor and sensory improvement, axon regeneration, inflammation reduction, an increase in tissue repair, and relief from neuropathic pain. Stem cells such as iPSCs can differentiate into any type of cell, MSCs lead to a significant improvement in neurological dysfunction, HSCs can be autologously-derived and have a record of safety, NSCs can replace damaged neural tissue, and transplanting HuCNS-SCs indicated consistent sensory improvements. The documented distinct adverse effects associated with stem cell transplantation include neuropathic pain, unusual sensations, muscle spasms, vomiting, and urinary tract infections. Several studies, clinical and preclinical, have established the safety and efficacy of stem cell therapy. Nevertheless, most studies are in preclinical phase I/II, requiring further studies to clarify this observation and to define their role in terms of functional recovery. As previously mentioned, there are still too many discrepancies that might be taken into account; proper guidelines focused on their correct management, including the identification of the most suitable stem cells, delivery site, and timing of intervention, should be considered. Hence, there are obstacles that must be overcome before stem cell therapy can become a frequently chosen treatment for spinal cord injuries, and the application of advanced biotechnological methods could resolve these obstacles. More research is required on the specific mechanisms of tissue repair and regeneration, more efficient delivery methods, differentiation and integration, and safety. Vital insights into the potential of stem cell therapy for SCI will be gained once the ongoing clinical studies are completed, and future research will contribute to the development of more effective treatments. Rigorous clinical trials (phase III/IV) with extending follow-up studies are required to confirm the advancements in stem cell therapy research. 

## Figures and Tables

**Table 1 brainsci-13-01697-t001:** List of clinical trials on stem cell therapy for SCI.

Identifier	Investigator	Title	Lesion Type	Cell Source	Study Phase	Effects on Neural Regeneration
NCT02481440	Li-Min Rong	Repeated Subarachnoid Administrations of hUC-MSCs in Treating SCI	Spinal cord injury	Human umbilical cord mesenchymal stem cells	Phase 2	hUC-MSCs is safe and effective, improved neurological dysfunction
NCT01909154	Jesus JV Vaquero Crespo	Safety study of local administration of autologous bone marrow stromal cells in chronic paraplegia (CME-LEM1)	Chronic paraplegia	Autologous bone marrow stromal cells	Phase 1	Motor enhancement, pain alteration, neurophysiological parameters improved
NCT01873547	Yihua An	Different Efficacy Between Rehabilitation Therapy and Stem Cells Transplantation in Patients with SCI in China (SCI-III)	Spinal cord Injury	Mesenchymal stem cells derived from umbilical cord	Phase 3	Not informed
NCT02574585	Ricardo Ribeiro-dos-Santos	Autologous mesenchymal stem cells transplantation in thoracolumbar chronic and complete spinal cord injury spinal cord injury	Thoracolumbar chronic SCI	Autologous bone marrow mesenchymal stem cells	Phase 2	Not informed
NCT02482194	Parvez Ahmed	Autologous mesenchymal stem cells transplantation for spinal cord injury-a phase I clinical study	Traumatic spinal cord injury at the thoracic level	Autologous BM MSCs	Completed	BMMSCs (intrathecal administration) is safe, no adverse events
NCT02981576	Abdalla Awidi	Safety and Effectiveness of BM-MSC vs. AT-MSC in the Treatment of SCI Patients.	Spinal cord injury	Bone marrow MSC (BM-MSC), adipose tissue MSC (AT-MSC)	Phase 2	Not informed
NCT01624779	Taehyeong Jo	Intrathecal transplantation of autologous adipose tissue derived MSCs in the patients with SCI	Clinical diagnosis of SCI	Adipose-derived MSCs	Phase 1	Neurological function improved (mild), no serious adverse events
NCT04288934	Fatima Jamali	Treatment of Spinal Cord Injuries With (AutoBM-MSCs) vs. (WJ-MSCs)	Spinal cord injury	AutoBM-MSCs, WJ-MSCs	Phase 1	Not informed
NCT01162915	Gabriel P. Lasala	Phase I, single center, trial to assess safety and tolerability of the intrathecal infusion of ex-vivo expanded bone marrow derived MSCs for the treatment of SCI	SCI clinical diagnosis (ASIA A)	Autologous bone marrow MSCs	Phase 1	Not informed
NCT03003364	Joan Vidal	Intrathecal Administration of Expanded Wharton’s Jelly Mesenchymal Stem Cells in Chronic Traumatic Spinal Cord Injury	Chronic traumatic spinal cord injury	Wharton’s jelly mesenchymal stem cells	Phase 2	Not informed
NCT01325103	Ricardo R. dos Santos	Phase I study of autologous bone marrow stem cell transplantation in patients with spinal cord injury	Spinal cord injury	Autologous bone marrow stem cell	Phase 1	Transplantation of autologous BMSCs is a feasible and safe technique
NCT01730183	Yashbir Dewan	Study the safety and efficacy of bone marrow derived autologous cells for treatment of SCI	Spinal cord injury	Bone-marrow-derived autologous cells	Phase 2	Not informed
NCT01274975	SangHan Kim	Autologous adipose derived MSCs transplantation in patient with SCI	Spinal cord injury	Adipose-derived MSCs	Phase 1	Intravenous administration of AD MSCs is safe with no adverse events
NCT01186679	Dr Arvind Bhateja	Surgical transplantation of autologous bone marrow stem cells with glial scar resection for patients of chronic SCI and intra-thecal injection for acute and subacute injury-a preliminary study	Chronic SCI	Autologous bone marrow stem cells	Phase 2	Not informed
NCT01393977	An Yihua	Difference between rehabilitation therapy and stem cells transplantation in patients with spinal cord injury in China	Spinal cord injury	Stem cells	Phase 2	Improved urinary control, muscle tension, motion, and self-care ability
NCT03308565	Mohamad Bydon	Adipose Stem Cells for Traumatic Spinal Cord Injury (CELLTOP)	Traumatic spinal cord injury	Adipose stem cells	Phase 1	Not informed
NCT02570932	Jesús JV Vaquero Crespo	Administration of Expanded Autologous Adult Bone Marrow Mesenchymal Cells in Established Chronic Spinal Cord Injuries	Chronic spinal cord injuries	Adult bone marrow mesenchymal cells	Phase 2	Neurological function improved (mild), no serious adverse events
NCT01186679	Dr Arvind Bhateja	Safety and Efficacy of Autologous Bone Marrow Stem Cells in Treating Spinal Cord Injury (ABMST-SCI)	Spinal cord injury	Autologous bone marrow stem cells	Phase 2	Not informed
NCT02481440	Min Li Rong	Repeated Subarachnoid Administrations of hUC-MSCs in Treating SCI	Spinal cord injury	Human umbilical cord mesenchymal stem cells	Phase 2	Improved control, motion, and self-care ability
NCT04331405	Vladimir A. Smirnov	Allogeneic Cord Blood Cells for Adults with Severe Acute Contusion Spinal Cord Injury	Severe acute contusion SCI	Allogeneic cord blood cells	Phase 2	Not informed
NCT04205019	Johannes P de Munter	Safety Stem Cells in Spinal Cord Injury (SSCiSCI)	Spinal cord injury	Neuro-cells	Phase 1	Not informed
NCT01769872	Tai-Hyoung Cho	Safety and Effect of Adipose Tissue Derived Mesenchymal Stem Cell Implantation in Patients with Spinal Cord Injury	Spinal cord injury	Mesenchymal stem cell	Phase 2	Not informed
NCT01321333	Stephen Huhn	Study of Human Central Nervous System Stem Cells (HuCNS-SC) in Patients with Thoracic Spinal Cord Injury	Thoracic spinal cord injury	Central nervous system stem cells	Phase 2	Not informed
NCT02163876	Stephen Huhn	Study of Human Central Nervous System (CNS) Stem Cell Transplantation in Cervical Spinal Cord Injury	Cervical spinal cord injury	Central nervous system stem cell	Phase 2	Not informed
NCT02152657	Ricardo R dos Santos	Evaluation of Autologous Mesenchymal Stem Cell Transplantation in Chronic Spinal Cord Injury: A Pilot Study	Chronic spinal cord injury	Mesenchymal stem cell	Phase 1	Not informed

## Data Availability

No new data were created or analyzed in this study.

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
