# Peer review of "An Insight into the Prospects and Drawbacks of Stem Cell Therapy for Spinal Cord Injuries: Ongoing Trials and Future Directions"

_brainsci, 2023, doi:10.3390/brainsci13121697_

Round 1
Reviewer 1 Report
Comments and Suggestions for Authors
The inefficiency of traditional methods of treating patients with spinal cord injury has prompted the development of alternative methods of treatment based on modern advances in cell biology and regenerative medicine, specifically the use of various types of stem/progenitor cells.
The authors aimed to compile evidence on the efficacy of cellular technologies in the treatment of spinal cord damage from clinical exploratory research in their review.
The authors partially answered this difficulty by demonstrating that the efficacy of cellular technologies in spinal cord damage is sadly low, but the reasons for this low efficacy were not provided.
The authors discuss the well-known facts of cell technologies' short-term or even ineffectiveness not only non the treatment of spinal cord injuries, but also in other conditions such as limb ischaemia, chronic heart failure, and so on.
The following deficiencies should be addressed in the work: 1) A lack of description of the design of the search for publications on the topic of the review - how the authors searched for papers; in what search engines; utilizing or not using programs that enable to automatically find papers by keywords, and so on.
2) The process for including and excluding publications from the review based on criteria such as cell type, mode of administration, frequency of administration, nosology, age, gender, country, and so on is not mentioned.
3) Only 30 papers from the list of citations were less than 5 years old.
About 30 reviews on the use of various types of stem cells and cell technology in spinal cord injury can be found in PubMed, including:
Abdolmohammadi K, Mahmoudi T, Alimohammadi M, Tahmasebi S, Zavvar M, Hashemi SM. Mesenchymal stem cell-based therapy as a new therapeutic approach for acute inflammation. Life Sci. 2023 Jan 1;312:121206. doi: 10.1016/j.lfs.2022.121206. Epub 2022 Nov 18. PMID: 36403645.
Du X, Amponsah AE, Kong D, He J, Ma Z, Ma J, Cui H. hiPSC-Neural Stem/Progenitor Cell Transplantation Therapy for Spinal Cord Injury. Curr Stem Cell Res Ther. 2023;18(4):487-498. doi: 10.2174/1574888X17666220509222520. PMID: 35538805.
Ou YC, Huang CC, Kao YL, Ho PC, Tsai KJ. Stem Cell Therapy in Spinal Cord Injury-Induced Neurogenic Lower Urinary Tract Dysfunction. Stem Cell Rev Rep. 2023 Aug;19(6):1691-1708. doi: 10.1007/s12015-023-10547-9. Epub 2023 Apr 28. PMID: 37115409.
Ribeiro BF, da Cruz BC, de Sousa BM, Correia PD, David N, Rocha C, Almeida RD, Ribeiro da Cunha M, Marques Baptista AA, Vieira SI. Cell therapies for spinal cord injury: a review of the clinical trials and cell-type therapeutic potential. Brain. 2023 Jul 3;146(7):2672-2693. doi: 10.1093/brain/awad047. Erratum in: Brain. 2023 Jun 27;: PMID: 36848323.
The review table is lengthy, uninformative, and largely refers to research without discussing the outcomes - in this form, it is pointless.
Author Response
Manuscript ID: brainsci-2740091
Title: An Insight into the Prospects and Drawbacks of Stem Cell Therapy for Spinal Cord Injuries: Ongoing Trials and Future Directions
Authors: Shahidul Islam Khan, Nazmin Ahmed, Kamrul Ahsan, Mahmud Abbasi, Rosario Maugeri, Dhiman Chowdhury, Lapo Bonosi, Lara Brunasso, Roberta Costanzo, Domenico Gerardo Iacopino, Giuseppe Emmanuele Umana, and Bipin Chaurasia
Dear Editor and Reviewer 1,
The authors of this manuscript are truly grateful for the time and the effort of the present reviewer’s suggestions. We firmly believe that all Your suggestions could enhance the quality and the effectiveness of the submitted manuscript. We tried, according to the Reviewer 1, to streamline the text and emphasized some points.
Please find below a point-by-point report of the manuscript corrections. All changes are visible through track changes modality on word doc attached.
- A lack of description of the design of the search for publications on the topic of the review- how the authors search for papers, in what search engines, utilizing or not using programs that enable to automatically find papers by keywords, and so on.
- Response: The methodology section has been added with required information suggested by the reviewers.
- The process for including and excluding publications from the review based on criteria such as cell type, mode of administration, frequency of administration, nosology, age, gender, country, and so on is not mentioned.
- Response: The revised version of the manuscript consists of a methodology section.
- Only 30 papers from the list of citations were less than 5 years old.
- Response: Well, there is a valid point regarding this. As it is a review article, we have collected article of a long rage of period. Our purpose is not to analyze the result of review articles, we have focused on clinical trials.

Reviewer 2 Report
Comments and Suggestions for Authors
This review manuscript, "An Insight into the Prospects and Drawbacks of Stem Cell Therapy for Spinal Cord Injuries: Ongoing Trials and Future Directions," offers a critical and balanced analysis of stem cell therapy's role in treating spinal cord injuries (SCI). It adeptly navigates through the complexities of SCI, the potential of various stem cells, and the intricacies of current clinical trials. The paper stands out for its comprehensive coverage of both the promising advances and the significant challenges in the field. It thoughtfully discusses the ethical and safety considerations, underscoring the need for more rigorous research and standardized protocols. Highlighting the trend towards combination therapies, the manuscript proposes innovative approaches for more effective treatments. Its conclusion is forward-looking, emphasizing the necessity of continued exploration and refinement in stem cell therapy for SCI. This review is a valuable contribution to the field, offering insights and guidance for future research.
To improve the quality of this manuscript, consider incorporating the following enhancements:
Main text:
1. The mention of psychological burden is important. Consider expanding this to include common psychological conditions faced by SCI patients, such as depression or anxiety, and their prevalence (Lines 44-45).
2. The etiology is well-described. Adding statistical data on the prevalence of traumatic vs. non-traumatic SCI could provide a clearer picture of their respective impacts (Lines 46-52).
3. The current treatment options are briefly mentioned. Expanding on why these treatments fail to result in significant functional improvement would add depth (Lines 58-62).
4. The regenerative strategies are briefly touched upon. A deeper explanation of how these strategies work, particularly the role of biomaterials, could be beneficial (Lines 63-65).
5. The two-stage damage mechanism in traumatic SCI is well-explained. Adding information about the time frame of these stages could provide more detailed insights (Lines 66-68).
6. The controversy regarding methylprednisolone use is crucial. It would be beneficial to briefly discuss alternative pharmacological interventions being explored due to this controversy (Lines 69-75).
7. While rehabilitative strategies are mentioned, elaborating on specific types of rehabilitation therapies and their efficacy would be informative (Lines 76-77).
8. The socioeconomic aspect is crucial but needs more detail. Discussing the disparities in access to care and rehabilitation services in different socioeconomic settings would be insightful (Lines 84-85).
9. The promise of stem cell therapy is highlighted. Providing a brief overview of different types of stem cells and their specific applications in SCI could add value (Lines 86-93).
10. The mention of numerous clinical trials is important. A discussion on the criteria for selecting patients for these trials, and the challenges faced in translating results from animal studies to human trials, would be beneficial (Lines 94-108).
11. The section clearly outlines the potential of stem cells in SCI treatment, highlighting motor and sensory function improvements. Including more recent studies or trials in this context would ensure the content is up to date (Lines 111-120).
12. The ethical concerns surrounding ESCs and the shift towards iPSCs are well-discussed. Elaborating on the specific techniques used to reprogram adult cells into iPSCs would add depth to this section (Lines 125-133).
13. The text provides valuable insights into clinical research outcomes using mesenchymal stromal cells. Including comparative effectiveness of these cells against other cell types in SCI treatment would be beneficial (Lines 135-147).
14. The diverse approaches in stem cell therapy are well-covered. Discussing the mechanisms by which these different approaches aid in SCI recovery could enhance understanding (Lines 148-163).
15. The role of hematopoietic stem cells in SCI is intriguing. A comparison of their efficacy with other stem cell types would provide a more comprehensive view (Lines 164-170).
16. The mention of neural stem/progenitor cells is important. Clarifying their specific advantages or limitations in SCI treatment compared to other stem cells could add value (Lines 171-177).
17. The challenges section is critical. Discussing potential solutions or ongoing research addressing these challenges would be informative (Lines 179-197).
18. The limitations in functional improvements and trial criteria are well-noted. Suggesting possible improvements in trial design to overcome these limitations could be helpful (Lines 199-206).
19. The issue of spontaneous recovery and the lack of control groups in trials is a key point. Recommendations for incorporating control groups in future trials would strengthen this section (Lines 207-214).
20. This section highlights important limitations in current research. Suggestions for standardizing research methodologies could enhance the quality of future studies (Lines 216-221).
21. The future research directions are well outlined. Emphasizing the importance of interdisciplinary approaches, involving bioengineering and neurology, could broaden the scope of future research suggestions (Lines 222-239).
22. The concerns about the safety of stem cell therapy, particularly regarding tumor growth, are significant. Discussing advancements in monitoring and mitigating these risks would be beneficial (Lines 240-245).
23. The prospect of combining stem cell therapy with other treatments is exciting. Elaborating on how these combinations could be optimized for better outcomes would be insightful (Lines 246-248).
Conclusion:
24. The conclusion effectively summarizes the potential of stem cell therapy in SCI. It might be beneficial to briefly recap key findings from the studies mentioned earlier in the text to reinforce the conclusion.
25. While the safety and efficacy of stem cell therapy are mentioned, emphasizing the need for more extensive clinical trials (beyond phase I/II) to confirm these findings would be helpful.
26. The mention of discrepancies and the need for proper guidelines is crucial. Specifying the types of discrepancies and suggesting areas where guidelines are most needed would add clarity.
27. The text rightly points out the obstacles in stem cell therapy. Discussing potential strategies to overcome these obstacles, such as advancements in biotechnology or improved delivery methods, would provide a more comprehensive outlook.
28. The text correctly identifies the role of future research. Suggesting specific areas for future research, such as combination therapies or new stem cell sources, would make this conclusion more impactful.
Abstract:
29. The abstract begins with a clear definition of SCI and its impact on quality of life. To strengthen this, include brief statistics or data on the incidence of SCI globally to contextualize its significance.
30. This section highlights the absence of effective treatments for SCI. It could be enhanced by briefly mentioning the current standard treatments and their limitations.
31. The focus on stem cells as a regenerative treatment is well-articulated. Clarifying the types of stem cells being explored (e.g., embryonic, mesenchymal) early in the abstract could provide a clearer foundation for the reader.
32. The abstract effectively describes how stem cells target SCI pathologies. Expanding slightly on the mechanisms of action, like how stem cells provide nutritional support or function as scaffolds, would be informative.
33. While the mention of minor benefits from single-cell treatments is important, briefly noting the specific types of benefits observed (e.g., sensory improvements, motor function) would add specificity.
34. The abstract concludes with the idea of combination treatments. Providing a brief example of such a combination (e.g., stem cells with physical therapy) could illustrate this point more concretely.
Comments on the Quality of English LanguageThe quality of English language used in this paper appears to be proficient, with clear and articulate expression of complex scientific ideas. The terminology is appropriate for the subject matter, and the sentence structure effectively conveys the nuances of stem cell therapy in spinal cord injury treatment. The manuscript maintains a professional tone suitable for an academic audience, demonstrating a thorough understanding of the topic. However, ensuring consistency in terms of style, grammar, and technical jargon throughout the paper would further strengthen its linguistic quality.
Author Response
Manuscript ID: brainsci-2740091
Title: An Insight into the Prospects and Drawbacks of Stem Cell Therapy for Spinal Cord Injuries: Ongoing Trials and Future Directions
Authors: Shahidul Islam Khan, Nazmin Ahmed, Kamrul Ahsan, Mahmud Abbasi, Rosario Maugeri, Dhiman Chowdhury, Lapo Bonosi, Lara Brunasso, Roberta Costanzo, Domenico Gerardo Iacopino, Giuseppe Emmanuele Umana, and Bipin Chaurasia
Dear Editor and Reviewer 2,
The authors of this manuscript are truly grateful for the time and the effort of the present reviewer’s suggestions. We firmly believe that all Your suggestions could enhance the quality and the effectiveness of the submitted manuscript. We tried, according to the Reviewer 2, to streamline the text and emphasized some points.
Please find below a point-by-point report of the manuscript corrections. All changes are visible through track changes modality on word doc attached.
- Comment No: 1. The mention of psychological burden is important. Consider expanding this to include common psychological conditions faced by SCI patients, such as depression or anxiety, and their prevalence (Lines 44-45).
- Response: The suggested information has been added to the manuscript.
- Comment No: 2. The etiology is well-described. Adding statistical data on the prevalence of traumatic vs. non-traumatic SCI could provide a clearer picture of their perspective impacts (Lines 46-52).
- Response: The suggested information has been added to the manuscript.
- Comment No: 3. The current treatment options are briefly mentioned. Expanding on why these treatments fail to result in significant functional improvement would add depth (lines 58-62).
- Response: The suggested information has been added to the manuscript.
- Comment No: 4. The regenerative strategies are briefly touched upon. A deeper explanation of how these strategies work, particularly the role of biomaterials, could be beneficial (Lines 63-65).
- Response: The information is available in the section no. 3 of this manuscript.
- Comment No: 5. The two-stage damage mechanism in traumatic SCI is well-explained. Adding information about the frame of these stages could provide more detailed insights (Lines 66-68).
- Response: The suggested information has been added to the manuscript.
- Comment No: 6. Response: The suggested information has been added to the manuscript.
- Comment No: 7. Response: The suggested information has been added to the manuscript.
- Comment No: 8. Response: The suggested information has been added to the manuscript.
- Comment No: 9. Response: The suggested information has been added in subsequent parts of the manuscript.
- Comment No: 10. Response: The suggested information has been added in subsequent parts of the manuscript.
- Comment No: 11. Response: We have tried to update the information with available most recent studies.
- Comment No: 12. Response: The suggested information has been added to the manuscript.
- Comment No: 13. Response: The comparison is already available in the manuscript.
- Comment No: 14. Response: This does not go with the aim of this manuscript. We are working on another study to highlight the mechanism.
- Comment No: 15. Response: The suggested information has been added to the manuscript.
- Comment No: 16. Response: The suggested information has been added to the manuscript.
- Comment No: 17. Response: The suggested information are already available in the manuscript.
- Comment No: 18. Response: The suggested text has been added to the manuscript.
- Comment No: 19. Response: The suggested information has been added to the manuscript.
- Comment No: 20. Response: The suggested information has been added to the manuscript.
- Comment No: 21. Response: The suggested information has been added to the manuscript.
- Comment No: 22. Response: The suggested information has been added to the manuscript.
- Comment No: 23. Response: The suggested information has been added to the manuscript.
- Comment No: 24. Response: The suggested information has been added to the manuscript.
- Comment No: 25. Response: The suggested information has been added to the manuscript.
- Comment No: 26. Response: The suggested information has been added to the manuscript.
- Comment No: 27. Response: The suggested information has been added to the manuscript.
- Comment No: 28. Response: The suggested information has been added to the manuscript.
- Comment No: 29. Response: The suggested information has been added to the manuscript.
- Comment No: 30. Response: The suggested information has been added to the manuscript.
- Comment No: 31. Response: The suggested information has been added to the manuscript.
- Comment No: 32. Response: The suggested information has been added to the manuscript.
- Comment No: 33. Response: The suggested information has been added to the manuscript.
- Comment No: 34. Response: The suggested information has been added to the manuscript.

Round 2
Reviewer 1 Report
Comments and Suggestions for Authors
Due to the limited efficacy of standard treatment modalities, the issue of clinical efficacy of different stem cells in the treatment of spinal cord injury remains relevant. This review article aims to bring together evidence from clinical research on the efficacy of cell therapy for spinal cord injury. Despite the corrections made in the initial edition of the article, the logic of including clinical trials in the second and third phases, where there is no data on the outcome of the clinical trial, in the article remains unclear.
It is possible that these are closed trials or that access to the trial results is restricted, but in this instance, the objectivity of their inclusion in the study is being questioned. It is difficult to understand the authors' logic when they state that it is possible to use this or that type of stem cell in the treatment of spinal cord injury yet do not show actual findings.
The writers' emotive message criticizing the high number of citations older than 5 years is most likely the result of a lack of understanding of the points required for reviewing articles and reviews - how much of the cited literature is new.
Author Response
Manuscript ID: brainsci-2740091
Title: An Insight into the Prospects and Drawbacks of Stem Cell Therapy for Spinal Cord Injuries: Ongoing Trials and Future Directions
Authors: Shahidul Islam Khan, Nazmin Ahmed, Kamrul Ahsan, Mahmud Abbasi, Rosario Maugeri, Dhiman Chowdhury, Lapo Bonosi, Lara Brunasso, Roberta Costanzo, Domenico Gerardo Iacopino, Giuseppe Emmanuele Umana, and Bipin Chaurasia
Dear Editor and Reviewer 1,
The authors of this manuscript are truly grateful for the time and the effort of the present reviewer’s suggestions. We firmly believe that all Your suggestions could enhance the quality and the effectiveness of the submitted manuscript.

Reviewer 2 Report
Comments and Suggestions for Authors
Thank you for the effort and time you have dedicated to revising this manuscript. It is evident that you have considered several of the comments provided. However, after reviewing the revised manuscript, I have observed that many key suggestions were not fully incorporated or addressed.
Please note that the conclusion section is just one example as follow, and similar attention to detail is needed throughout the manuscript.
1. Summary of Key Findings (Comment 24 from the First Round of Review): The conclusion summarizes the potential of stem cell therapy in SCI, but it lacks specific references to key findings from earlier studies. I recommend enhancing the conclusion by directly linking it back to pivotal results and conclusions from these studies, thus providing a cohesive and comprehensive summary of the manuscript’s findings.
2. Need for More Extensive Clinical Trials (Comment 25 from the First Round of Review): While the text acknowledges the need for further studies, it does not sufficiently emphasize the critical necessity of advancing beyond phase I/II trials. It would be beneficial to explicitly highlight the importance of progressing to later-stage clinical trials (phase III/IV) to underscore the urgency and significance of this advancement in stem cell therapy research.
3. Discrepancies and Guidelines (Comment 26 from the First Round of Review): The manuscript mentions discrepancies and the need for guidelines but falls short in specifying the nature of these discrepancies and the areas where guidelines are most needed. Detailed examples of these discrepancies, along with suggestions for specific areas where guidelines are crucial, would greatly enhance this section of the text.
4. Strategies to Overcome Obstacles (Comment 27 from the First Round of Review): The conclusion mentions using advanced biotechnological methods to overcome obstacles, but it lacks an in-depth discussion of potential strategies. Expanding on this point by discussing specific advancements in biotechnology, improved delivery methods, or other innovative strategies could significantly enhance the understanding of how these obstacles might be overcome in stem cell therapy.
5. Specific Areas for Future Research (Comment 28 from the First Round of Review): The conclusion identifies the need for future research but does not detail specific areas like combination therapies or new stem cell sources. Outlining specific promising research areas, such as investigating new sources of stem cells or exploring combination therapies, would provide clear direction for future research and add substantial value to the conclusion.
In summary, while appreciable progress has been made, the manuscript would benefit from a more thorough incorporation of the provided feedback, particularly in detailing and expanding on certain points.
In addition, I recommend that the rebuttal letter be written more meticulously. It should clearly detail what sections have been revised, what remains unaltered, and if there are any areas that could not be modified due to certain difficulties. This clarity will greatly aid in understanding the changes made and the reasons behind any unaddressed comments.
Furthermore, I recommend that future rebuttal letters avoid generic statements like "The suggested information has been added to the manuscript" unless the changes accurately reflect the requested modifications. A clear explanation of how each comment was addressed, or a justification for any comment that could not be incorporated, would greatly enhance the revision process and facilitate a more productive dialogue.
At this stage, I encourage a more detailed revision that aligns closely with the feedback provided.
Comments on the Quality of English LanguageMinor editing of English language required
Author Response
Manuscript ID: brainsci-2740091
Title: An Insight into the Prospects and Drawbacks of Stem Cell Therapy for Spinal Cord Injuries: Ongoing Trials and Future Directions
Authors: Shahidul Islam Khan, Nazmin Ahmed, Kamrul Ahsan, Mahmud Abbasi, Rosario Maugeri, Dhiman Chowdhury, Lapo Bonosi, Lara Brunasso, Roberta Costanzo, Domenico Gerardo Iacopino, Giuseppe Emmanuele Umana, and Bipin Chaurasia
Dear Editor and Reviewer 2,
The authors of this manuscript are truly grateful for the time and the effort of the present reviewer’s suggestions. We firmly believe that all Your suggestions could enhance the quality and the effectiveness of the submitted manuscript. We tried, according to the Reviewer 2, to streamline the text and emphasized some points.
Please find below a point-by-point report of the manuscript corrections. All changes are visible through track changes modality on word doc attached.
- The conclusion has been significantly revised.
- The relevant text in the conclusion section have been revised.
- Please, refer to the paragraph 5 of future research section.
- Paragraph 6 of the future research section combinedly addresses the comments no 4,5.
